# Porcine Cortical Bone Lamina as a Predictable Technique for Guided Bone Regeneration: Histomorphometric and Radiographic Evaluation

**Michele Antonio Lopez** [1,*,†] **, Pier Carmine Passarelli** [1,†] **, Andrea Netti** [1] **, Antonio D'Addona** [1] **, Francesco Carinci** [2] **, Piotr Wychowański** [3] **and Francesco Cecchetti** [4]

1   Department of Head and Neck and Sensory Organs, Division of Oral Surgery and Implantology, Institute of Clinical Dentistry, Gemelli Foundation for the University Policlinic, Catholic University of the Sacred Heart, 00168 Rome, Italy
2   Department of Translational Medicine, University of Ferrara, 44121 Ferrara, Italy
3   Department of Oral Surgery, Medical University of Warsaw, 6 St. Binieckiego Street, 02-097 Warsaw, Poland
4   Department of Social Dentistry and Gnathological Rehabilitation, National Institute for Health, Migration and Poverty (NIHMP), 00153 Rome, Italy
*   Correspondence: micheleantonio.lopez@gmail.com
†   These authors contributed equally to this work.

**Abstract:** The stability of bone regenerated through Guided Bone Regeneration (GBR) around implants is crucial for long-term success. In this case series, changes in marginal bone levels (MBL) around implants placed in a regenerated bone using heterologous cortical lamina technique were radiographically measured. In addition, bone samples were obtained and submitted to histological and histomorphometric analysis. Thirty implants were placed in regenerated bone sites 8 months after the regenerative surgery; in the same surgical stage, a hard tissue biopsy was taken using a trephine bur and submitted to histologic and histomorphometric analysis. Changes in the marginal bone level, mesial and distal to the implant shoulder, were measured between prosthetic loading and the last follow-up, 2 years later. No implants were lost, and all could be deemed successful at the last follow-up. Only a minimal mean variation in the position of the marginal bone level was observed, both at the mesial ($0.11 \pm 0.49$ mm) and at the distal level ($0.03 \pm 0.19$ mm). The bone lamina had been resorbed after 8 months, and new bone had developed in close connection to the biomaterial. The average percentage of newly formed bone was 28%, while only 10% of the samples were composed of residual biomaterial; bone marrow and connective tissue composed the remaining part of the samples. This regeneration technique allowed, thanks to the rigidity of the lamina, the regeneration of new bone, which is stable after the prosthetic load. Further studies are needed to compare this procedure with those adopting non-resorbable, titanium-supported membranes.

**Keywords:** heterologous cortical lamina; bone regeneration; implant prostheses; resorbable biomaterials; dentistry; dental implants

## 1. Introduction

Modern implantology techniques have reported success rates of over 95% in various studies; implants are, therefore, a predictable treatment option and a viable alternative to classical prosthetic techniques for patients who have lost one or more teeth [1].

Periodontal disease, extractions, or traumas, can lead to alveolar bone resorption, frequently hesitating in extensive bone atrophies [2]. In order to insert the implant according to the modern concept of prosthetically guided implantology, these atrophies have to be treated to achieve a healthy bone volume that can allow a proper inter-arch relationship and proper implant positioning in the different planes of space [3]. Different types of bone grafts can be adopted: they can either be of autologous, homologous, or heterologous origin.



Autologous bone has osteoinductive, osteosynthetic, and osteoconductive capabilities and has long been defined as the "gold standard" in guided bone regeneration (GBR) [4]. Donor sites can either be intra-oral or extra-oral; however, as two surgical accesses are needed, this procedure causes considerable discomfort to the patient and increases morbidity [4]. Heterologous bone grafting requires only one surgical access, significantly reducing the patient's post-operative discomfort and morbidity; on the other hand, the efficacy of this type of material is lower than that of autologous bone due to the lack of osteoinductive and osteogenic properties [2]. Some authors claim that the performance of the two materials overlaps in some clinical situations [5]. In contrast, other authors suggest that using a mix of heterologous and autologous can maintain the latter's properties while decreasing the amount needed [6]. Several techniques can be adopted to regenerate lost bone volumes [7–10], and different biomaterials are commonly adopted [11,12]. Over time, several GBR techniques have been proposed in which stability and absence of micromovements are necessary conditions to promote revascularization and graft integration [13,14]. If these conditions are lacking, the grafted bone may become necrotic or infected, compromising the success of the surgery [13]. For these reasons, when resorbable membranes are adopted, a lower success rate than non-reasorbable membranes is reported in the literature [15]. Recently, new resorbable, porcine heterologous cortical bone lamina was introduced to the market, and some authors have applied them to the regeneration of bone defects to restore the bone volumes needed for placing implants [16,17].

After implants insertion, assessment of the marginal bone level (MBL) and clinical data allow the health of the implant to be monitored and the onset of peri-implant disease (mucositis and peri-implantitis) to be adequately intercepted [18]. Hämmerle et al., in a recent systematic review, have shown no statistically significant differences between the native and regenerated bone in terms of implant survival and success [19].

The aim of this study is to radiographically assess the variation of marginal bone levels around implants inserted in regenerated sites after prosthetic loading and to provide a histomorphometric evaluation of the regenerated tissues. Bone regeneration was performed using a GBR technique with porcine heterologous biomaterials: particulate bone stabilized by a cortical bone lamina and covered by a resorbable collagen membrane.

## 2. Materials and Methods

This study is a case series in which data from 20 patients admitted to the Oral Surgery Unit of the Agostino Gemelli University Hospital (Rome, Italy) between January 2016 and April 2016 were retrospectively analyzed. There were 9 females and 11 males, with a minimum age of 27 years and a maximum age of 75 years. These patients presented a partial edentulism, 1 to 3 missing teeth in the posterior sextants of the mandible, treated with GBR, using a porcine heterologous cortical bone lamina, followed by the insertion of one or two implants to support a fixed restoration.

The clinical criteria for patients' exclusion from surgical treatment were medical and/or general contraindications for the surgical procedures (ASA score $\geq$ 3), acute oral infections, uncontrolled periodontal diseases, ongoing bisphosphonate therapy, history of chemotherapy or radiation therapy in the head or neck region, immunocompromised status, uncontrolled diabetes mellitus, systemic disease that could compromise post-operative healing and osseointegration, pregnancy or lactating, alcohol or drug abuse, smoking more than 10 cigarettes per day (smoking patients are asked to quit smoking one week before surgery and refraining from smoking for the next three weeks), psychiatric illness, lack of motivation to undergoing follow up period.

The study was authorized by the Ethics Committee of the "Fondazione Policlinico Universitario A. Gemelli", with protocol number "0009738/22".

### 2.1. Pre-Surgical Phase

Before the surgical procedure, any periodontal defect was treated so that no pocket with a PPD $\geq$ 5 mm was present. The treatments included four patients who had been

classified for mild, localized forms of adult periodontitis (classification until 2017), treated as described in the previous 12 months and who were stabilized at the time of re-evaluation for inclusion with no probing >4 mm. All patients underwent a preliminary preparatory phase including mechanical debridement with ultrasonic and manual instruments and had to maintain a full mouth plaque score (FMPS) and full mouth bleeding score (FMBS) ≤ 20%. Subjects were also instructed on the oral hygiene procedures they had to follow immediately before and after the surgical procedures. A basic preoperative CBCT scan of the areas of bone to be regenerated was performed (Pax-i3D Smart, 50–99 kVp/4–16 mA; Vatech, Hwaseong, Korea). Data on the subjects' age, gender, and dental status was obtained, and their dental history was updated immediately before the surgery.

*2.2. Surgical Procedure*

The patient underwent an antibiotic prophylaxis 1 h before surgery with 2 g amoxicillin + clavulanic acid in tablets (Augmentin 875 mg/125 mg, GSK, Brentford, UK) and a 1-min 0.20% chlorhexidine digluconate mouthwash, immediately before the beginning of the surgery.

After the infiltration of a local anesthetic (mepivacaine 20 mg/mL + adrenaline 1: 100,000, Pierrel S.p.A., Capua, Italy), a crestal incision was made in the edentulous area, continuing with an intrasulcular incision on the closest adjacent teeth on both sides of the defect. Full-thickness vestibular and lingual flaps were raised, incorporating the interdental papilla of adjacent teeth in the vestibular flap. The bone surfaces were freed of soft tissues, and a series of perforations were made in the cortical bone of the edentulous area with a round bur on a surgical handpiece to promote bleeding.

Bone defects were measured with a graduated periodontal probe to shape a porcine-derived cortical bone lamina (Lamina, Osteobiol, Tecnoss Dental S.R.L, Torino, Italy) of adequate size. The laminas were fixed through titanium osteosynthesis screws (Q-Bone Trinon, ROEN, Torino, Italy) positioned apically to the bone defect to reform a wall and thus create a containing healing box connecting the mesial and distal bone peaks (Figure 1). The newly formed box was filled with a heterologous biomaterial consisting of prehydrated and collagenated cortical-spongiosus granules with a size of 600–1000 μm (mp3 Osteobiol), and covered with a suitably shaped resorbable collagen membrane (Evolution Osteobiol, thickness 0.5 ± 0.1 mm) (Figures 2 and 3). Flaps were sutured (Supramid 4/0, Braun, Milano, Italy) using simple detached points, and post-operative radiographs were taken (Figure 4).

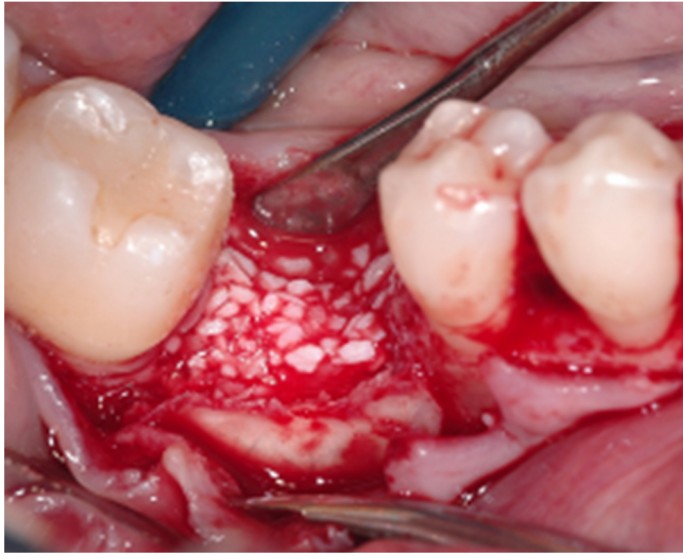

**Figure 1.** Creation of the containing "box" using a heterologous cortical lamina.

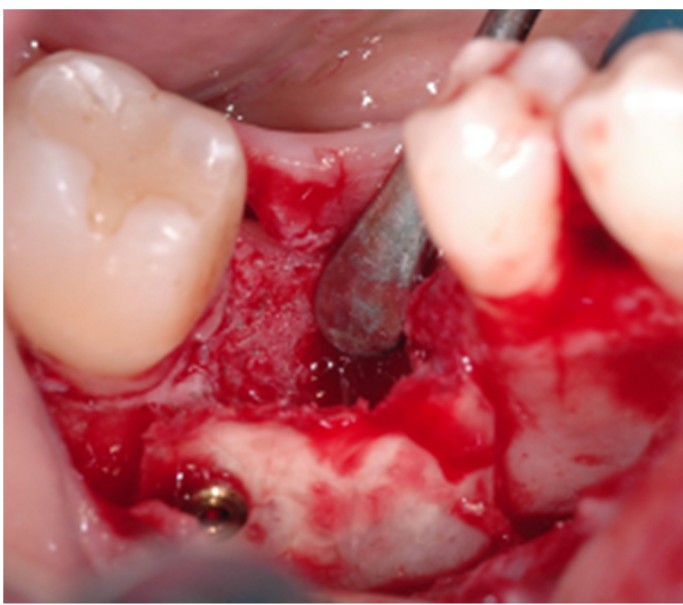

**Figure 2.** Filling the "box" with collagenated heterologous bone particulates.

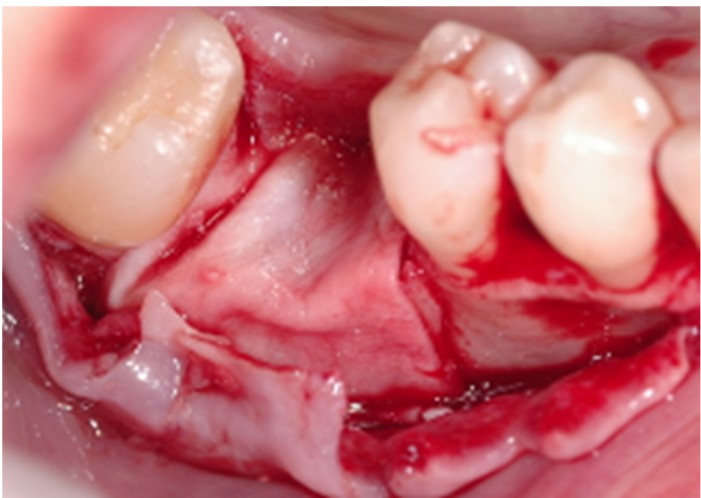

**Figure 3.** Placement of a collagen membrane to protect the graft.

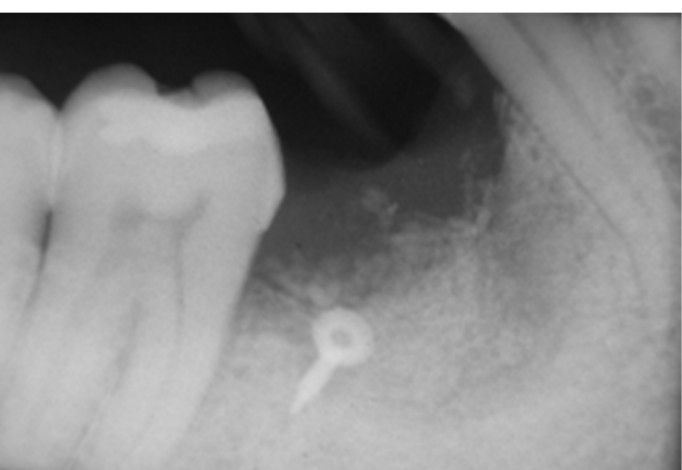

**Figure 4.** Post-operative Rx showing bone defect correction.

After 8 months, implants were inserted (Adapta Uniqo, FMD Srl, Roma, Italy) following a submerged protocol by elevating a small flap composed of a central crestal incision and two vestibular releasing incisions. Implants were placed juxta crestally following the steps described in the drilling protocol provided by the implant manufacturer for the type of bone encountered, reaching a final insertion torque corresponding to 40 N. Simultaneously, hard tissue biopsies were taken using a 4-mm-diameter trephine burr, and detached sutures were applied to obtain first intention healing. A unique identification number was assigned to each biopsy specimen.

After 3 more months, implants were exposed, and healing screws were inserted, remaining in place for 15 days. Then, at the end of an overall period of 4 months after implant placement, the impression procedures were performed (Impregum medium body polyether, 3M Italia Srl, Milano, Italy), and implants were loaded with cement-retained, veneered zirconia crowns. The prosthetic abutment is based on the principle of platform switching with conical connection associated with an antirotational hexagon (Adapta Uniqo).

Patients were monitored periodically every 4–6 months during the follow-up period. They underwent professional oral hygiene to maintain FMPS and FMBS ≤20% and to follow instructions for oral hygiene at home by adequate brushing, flossing and mouthwash with chlorhexidine 0.12%, if necessary.

### 2.3. Peri-Implant Bone Levels Assessment

Intra-oral radiographs were taken immediately after delivery of the prosthesis and 2 years after. The radiographic evaluation was performed in single-blind by an additional examiner (P.C.P.). Before evaluating the radiographs, the examiner was calibrated following defined instructions and reference radiographs with differing MBL measurements. The radiographs were taken using the parallel long cone technique and Rinn Radiograph Holders, with the sensor positioned perpendicular to the implant fixture, using a silicone check on the biteblock to standardize the procedure. These silicone checks were stored for each patient and used for subsequent radiographic controls. The magnification factor was measured on each radiograph by dividing the known diameter of the implant by the diameter measured on the radiograph. MBLs were assessed by defining 2 specific landmarks on the radiographs of each patient: The implant shoulder (IS) and the most coronal point of crestal bone (CB) in contact with the implant. A line parallel to the axis of the implant was drawn between CB and IS (CB-IS). Measures on mesial (MBLm) and distal (MBLd) aspects were obtained (Figure 5). The magnification factor subsequently adjusted these values. Then, we measured the bone resorption after the implant loading until the 2 years follow-up appointment. All measurements were taken using the DBSWIN 5 software (Durr Dental, Bietigheim-Bissingem, Germany).

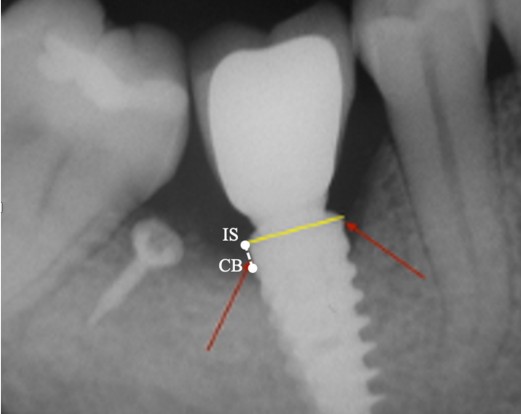

**Figure 5.** The implant is prosthesized 4 months after its insertion and bone levels around the implant are measured from the implant shoulder (yellow line) to the most coronal contact point of the mesial and distal marginal ridge (points are indicated by arrows in red). Legend: IS: implant shoulder; CB: crestal bone.

*2.4. Histologic and Histomorphometric Analysis*

Twenty samples, one per patient, were taken with a 4-mm-diameter trephine drill at the implant insertion site (in the case of insertion of two implants from the distal site), so as not to affect patient morbidity. Bone biopsies were fixed in 10% phosphate-buffered formalin, followed by decalcification in a hydrochloric acid/formic acid solution (4/5%). After decalcification, samples were dehydrated in a series of alcohol baths and then embedded in paraffin. The 5-μm sections were obtained on the long axis of the sample, taking portions close to but not including native bone and then, were prepared and stained with hematoxylin/eosin (h&e). The slides were then subjected to digital scanning at various magnifications to evaluate the presence and characteristics of the newly formed bone, of the remaining grafted material, and the integration of the grafted material with the surrounding tissues. The images from each area of the biopsy core were obtained and analyzed using image analysis software (ImageJ, NIH, http://rsb.info.nih.gov/nih-image, accessed on 20 January 2017) [20]. The image, via the 'Image' and 'Type' commands, was converted from 'RGB-color' to '8-bit' format, and then an appropriate threshold range was chosen to make the section stand out against the white background of the image with the 'Adjust' and 'Threshold' commands. Then the section area of the was calculated using 'analyze' and 'Measure'. Once the total area was calculated, the 'Freehand Selections' command was used to select the regions of residual biomaterial (RB), newly formed bone (NFB), and soft tissue (ST), and the section area was calculated using 'Analyze' and 'Measure' (Figure A1). For each sample, the areas of the different tissues and the total area of the micrograph were measured in pixels, the latter subtracted from possible empty areas derived from histological processing, thus allowing percentages, with an Excel spreadsheet.

## 3. Results

Twenty patients who underwent this procedure were included; 20 bone defects were regenerated, in the vertical and horizontal dimensions, and 30 implants were inserted. The implants' diameter varied from 3 mm to 4.5 mm, and their length varied from 8 mm to 12 mm. All implants were equipped with sandblasted, large grit, and acid-etched (SLA) surface up to the implant neck. There have been inserted 12 single-unit implant prostheses (12 implants) (40%), 7 three-unit bridge implant prostheses (14 implants) 46.7%, 2 two-unit bridge implant prostheses (4 implants) 13.3%. After the prosthetic finalization, these patients were followed for two years. The mean change in bone position at the mesial and distal site between the radiographs taken immediately after implant loading, and the radiographs taken at the final follow-up appointment are presented in Table 1. It can be stated that, overall, in 21 implants, the marginal bone was stable, as no change was observed either at the mesial or at the distal site at the final follow-up appointment, while in the remaining 9 implants, resorption was observed. The mean resorption observed was 0.11 ± 0.49 mm (mean + standard deviation) at the mesial site and 0.03 ± 0.19 mm (mean + standard deviation) at the distal site.

**Table 1.** The mean change in bone position at the mesial and distal site, between the radiographs taken after implant loading, and at the final follow-up appointment.

| Variable (mm) | Mean | Std dev | Std Err | 95% CI |
|---|---|---|---|---|
| MBLm | 0.11 | 0.49 | 0.16 | −0.46/0.24 |
| MBLd | 0.03 | 0.19 | 0.06 | −0.17/0.11 |

MBLm = Mesial marginal bone level, MBLd = Distal marginal bone level Std dev = standard deviation; Std err = standard error, CI = confidence interval.

*Histologic and Histomorphometric Results*

New bone formation was evaluated: in most subjects, the granules were largely resorbed and constituted only a scant part of the defect and were confined within its boundaries; the lamina was completely resorbed in all subjects and could not be observed in any of them. Bone trabeculae showed a lamellar structure with different thicknesses.

Histological examination showed newly formed bone interconnecting residual biomaterial, in close contact with the residual graft particles, indicating the osteoconductive capabilities. (Figures 6 and 7) The amount of bone formation was greater in the area closer to the native alveolar bone, when compared to the most coronal portions of the defects. All specimens displayed no appearance of inflammation or infection induced by the biomaterials. The average percentage of newly formed bone was 28.27% ± 10.26, while the percentage of residual biomaterial was 10.39% ± 11.60, while bone marrow and connective tissue composed the remaining part of the samples. The histomorphometric variables are presented in Table 2.

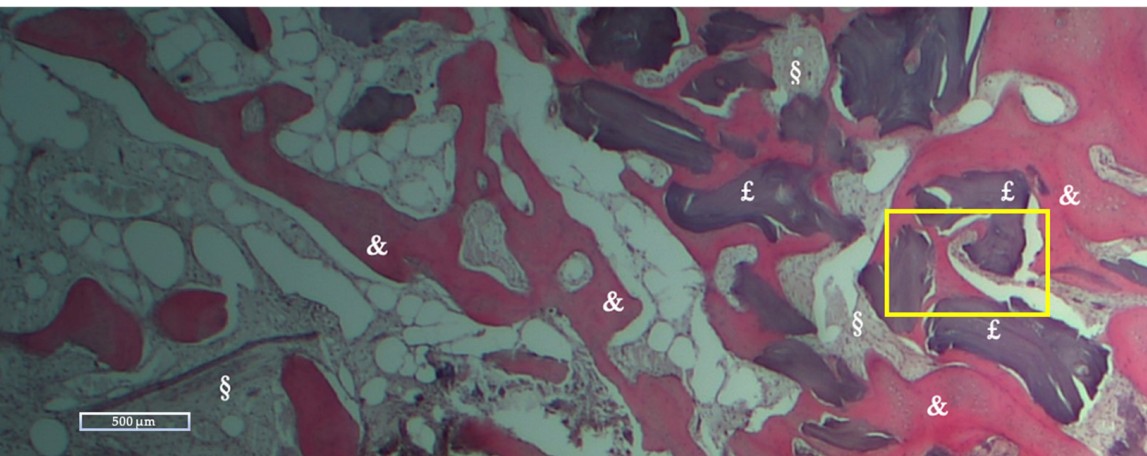

**Figure 6.** The regenerated tissues 8 months after the procedure stained with h&e. Graft particles are interconnected by bridges of newly formed bone (scale bar = 500 µm). Legend: £ Residual biomaterial, & Vital bone, § Soft tissue. Yellow rectangle, magnified in Figure 7.

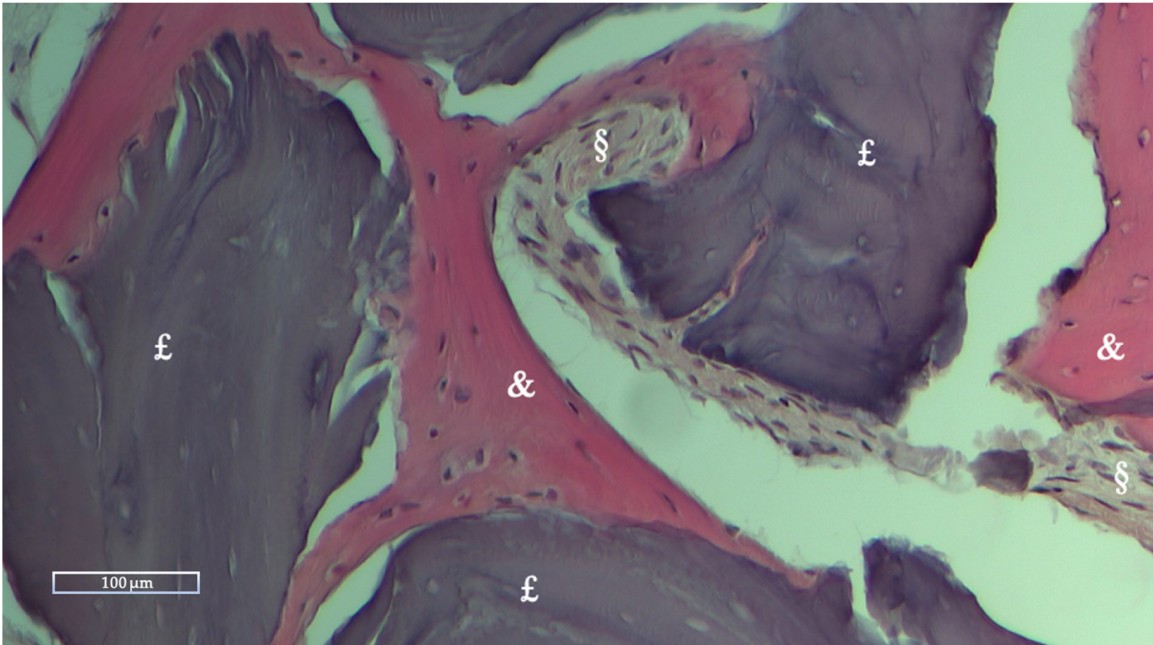

**Figure 7.** Higher magnification detail of Figure 6. The regenerated tissues 8 months after the procedure stained with h&e. Close contact between graft particles and newly formed bone. Legend: £ Residual biomaterial, & Vital bone, § Soft tissue (scale bar = 100 µm).

**Table 2.** Results of histomorphometric analysis from images of each biopsy core area taken at the time when the implant site was prepared.

|  | NFB | RB | SF |
|---|---|---|---|
| Mean % ± Std dev | 28.27 ± 10.62 | 10.39 ± 12.01 | 61.33 ± 12.08 |
| Range % (Min-Max) | 16.51–59.74 | 0–36.89 | 40.26–75.59 |

NFB = Newly Formed bone, RB = Residual Biomaterial, SF = Soft tissues, Std dev = standard deviation.

## 4. Discussion

This study aimed to evaluate the stability of the regenerated bone obtained with this technique, involving the application of a porcine bone lamina and bone particulate: only 9 of the 30 implants revealed minimal marginal bone loss. In addition, implants inserted in the regenerated bone bore loading of zirconia single crowns up to three elements fixed-partial-dentures, and all implants survived after a follow-up of 2 years. These data agree with the systematic review by Hämmerle et al. that there are no statistically significant differences between the native and regenerated bone in terms of survival and implant success [19]. The results found seem to reveal bone laminas as a valid and stable method to treat horizontal and vertical bone defects, in line with what the previous limited literature has stated: only a few studies have evaluated the clinical and histological outcome of this biomaterial. Some authors of the current study used the same protocol in a parallel case study, except for micronized filling (200–300 μm), but revealed faster resorption and reduced graft stability [16]. Pagliani et al. [21] have reported a success rate of 94.7% among several regenerative procedures in which Lamina was adopted and, after a 1-year follow-up, all but one of the implants placed in the regenerated defects survived, with a survival rate of 97.1%. Only a few previous studies have provided a histological analysis of GBR performed with bone laminas, and this is the first to have conducted a histomorphometric analysis. The results are in line with previous articles [21,22] in which, after a healing time between 6 and 12 months, new bone was formed in the apical extension of the defect, while, in the coronal extension, melted and highly vascularized connective tissue was formed. More mineralized bone was observed in the apical portion of the defect compared to its coronal extension. This finding could be related to the porosity of the membrane that allowed the growth of vascular cells, which is also frequent in other regenerative procedures employing non-resorbable membranes [23]. The formation of new vessels and the absence of foreign body reactions demonstrate that the lamina is biocompatible. Moreover, in the study of Rossi et al., at 6 months after GBR the lamina had not been completely resorbed but replaced primarily by cortical bone [22]. At the time of implant placement, 8 months after the regenerative procedure, the lamina was undistinguishable from the native bone and had been entirely resorbed. Therefore, as revealed by the histomorphometric analysis, the stability allowed the osteoconductive biomaterial to fill the defect with new sound bone. The slow resorption time of these membranes makes them suitable for vertical and horizontal regenerative procedures; while resorbable collagen membranes are rapidly resorbed by the host's immune system [24,25], the lamina can support the flap and protects the graft for a longer time. While no study has defined the resorption time needed for resorbable membranes, a barrier effect of up to 6 months is desirable, and many collagen barriers fail to achieve this objective [26]. On the other hand, non-resorbable membranes are more challenging to use and, while particularly efficient in vertical bone defects [27,28], are associated with a higher risk of complications, such as flap dehiscence, membrane exposure, and membrane infection, which can compromise the entire procedure if not adequately treated [29,30].

Patients were monitored and treated during the follow-up to maintain FMPS and FMBS ≤ 20%. This is a routine practice in our clinic to control the occurrence of mucositis and peri-implantitis, but it allowed us to keep the sample and, thus, the data collection homogeneous.

In addition, all patients received the same antibiotic prophylaxis as recommended in GBR procedures where significant oral bleeding and exposure to potentially contaminated tissue occur [31]. However, antibiotics' administration and effect are controversial and do not appear to improve or change post-surgical outcomes [32]. While long-term survival and implant success are influenced by a customized care program, which includes the removal of biofilm from implants, performed by professionals and during home oral hygiene, to control bacterial load and the oral microbiome [18].

Two years after prosthetic loading, the regenerated bone was stable, and implants were all successful, showing that the application of a heterologous bone lamina covered by a resorbable membrane, is an efficient substitute for traditionally used non-resorbable membrane.

The limitations of this case series relate primarily to the lack of a control group involving biomaterials or alternative techniques of proven efficacy to compare the results obtained with Lamina. Moreover, by setting up the study as an RCT and with a larger sample, statistical differences could have been appreciated, conferring strength to the results.

Considering some procedural limitations, it would be more comprehensive for biopsies to be taken at different times to properly assess the physiological processes that lead to the resorption of lamina and biomaterial and allow the deposition of new healthy bone. Moreover, the radiological examination was conducted only by two-dimensional radiographs, which cannot assess the volumetric changes after bone regeneration.

## 5. Conclusions

This regeneration technique allowed, thanks to the rigidity of the Lamina, the regeneration of new bone, which is stable after the prosthetic load.

The mean change in bone level, between the radiographs taken after implant loading and at the final follow-up appointment was $0.11 \pm 0.49$ mm at the mesial site and $0.03 \pm 0.19$ mm at the distal site. The bone lamina had been resorbed after 8 months, and new bone had developed in close connection to the residual biomaterial. The average percentage of newly formed bone was $28.27\% \pm 10.26$, while the percentage of residual biomaterial was $10.39\% \pm 11.60$, and bone marrow and connective tissue composed the remaining part of the samples ($61.33\% \pm 12.08$).

Further research is needed to compare this procedure with those adopting non-resorbable, titanium-supported membranes. Finally, more studies, with control group, more patients and longer follow-ups are needed to confirm present findings and improve the data obtained.

**Author Contributions:** Conceptualization, M.A.L. and P.C.P.; Methodology, A.D. and M.A.L.; Software, P.C.P.; Validation, A.D. and F.C. (Francesco Carinci); Formal Analysis, A.N., F.C. (Francesco Cecchetti); Investigation, M.A.L. and P.C.P.; Resources, A.D. and M.A.L.; Data Curation P.C.P.; Writing—Original Draft Preparation, M.A.L. and P.C.P.; Writing—Review & Editing, A.N., F.C. (Francesco Cecchetti), P.W.; Visualization, A.N., P.W. and F.C. (Francesco Carinci); Supervision, A.N., P.W. and F.C. (Francesco Carinci); Project Administration, M.A.L. and A.D.; Funding Acquisition, A.D. and M.A.L. All authors have read and agreed to the published version of the manuscript.

**Funding:** This research received no external funding.

**Institutional Review Board Statement:** The study was conducted according to the guidelines of the Declaration of Helsinki of 1975, as revised in 2013. The study was authorized by the Ethics Committee/Institutional Review Board of the "Fondazione Policlinico Universitario A. Gemelli", with protocol number "0009738/22".

**Informed Consent Statement:** All subjects involved in the study obtained informed consent for the clinical treatment they underwent, according to the criteria of good clinical practice. All participants were fully informed about the guarantee of anonymity, the reason for the research, and the use of their data.

**Data Availability Statement:** The data presented in this study are available on request from the corresponding author.

**Conflicts of Interest:** The authors declare no conflict of interest.

**Appendix A**

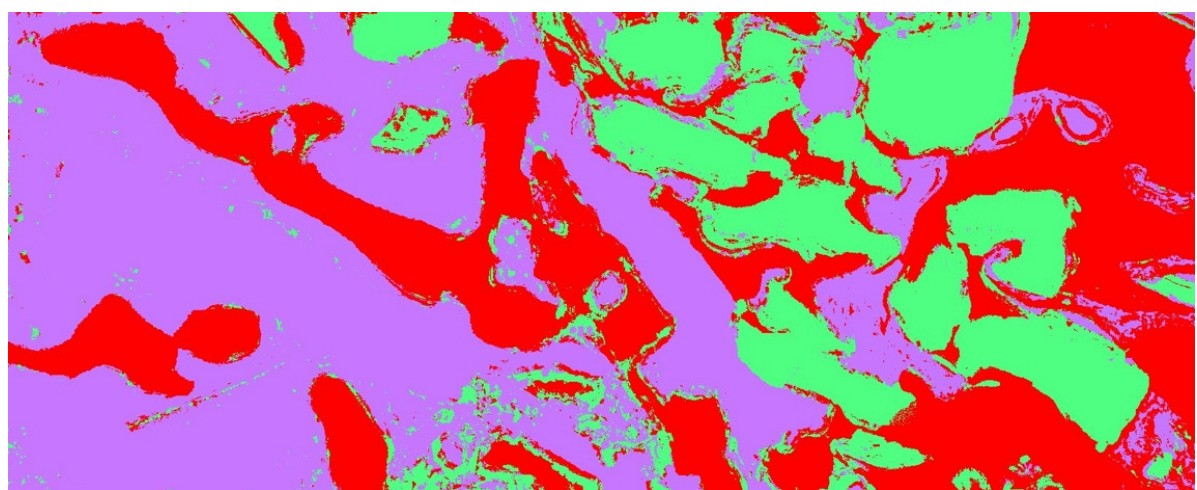

**Figure A1.** Segmentation of the histological image in Figure 6 with ImageJ software (NIH, http://rsb.info.nih.gov/nih-image, accessed on 20 January 2017), using different colors. Green: residual biomaterial; red: newly formed bone; lilac: soft tissue.

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
