# Peer review of "Porcine Cortical Bone Lamina as a Predictable Technique for Guided Bone Regeneration: Histomorphometric and Radiographic Evaluation"

_applsci, doi:10.3390/app122010285_

Round 1

Reviewer 1 Report

A clinical study is always welcome and would be of interest to the readers of the MDPI Journals.

Manuscript titled „Porcine cortical bone lamina as a predictable technique for 2 Guided Bone Regeneration: histomorphometric analysis after 8 months and radiographic evaluation after 2- years of follow-up. Case series” is original. The results are interpreted appropriately. The article is written in an appropriate way.

The treatment was performed with the highest technical standards. The methods and tools are described with sufficient details to allow another researcher to reproduce the results.

The conclusions are interesting for the readership of the Journal. The paper would be attractive for a wide readership.

Publishing this article is for the benefit of overall. It provides an advance towards the current knowledge. 

Author Response

Dear Reviewer,

We thank you for your work in reviewing our manuscript. We greatly appreciate your comments on our research work and hope that it can be published and reach a wide audience.

Best regards

Reviewer 2 Report

Dear Authors,

I have reviewed with interest your work entitled “Porcine cortical bone lamina as a predictable technique for Guided Bone Regeneration: histomorphometric analysis after 8 months and radiographic evaluation after 2- years of follow-up. Case series”.

It is a well-documented work; however, some revisions are required for acceptance in the Applied Sciences Journal.

Please, provide a point-by-point response to all Reviewers’ comments and highlight the changes with a different color mark for each reviewer, so that modifications can be easily found among the manuscript. 

Thank you.

Title

I believe that the title should not contain “Case series”. This is a research article, but this should not be written in the title. 

Keywords: Being Applied Sciences a multidisciplinary Journal, keywords like “dentistry” and “dental implants” should be added.

Introduction

·      Lines 49-57: other references should be added in this paragraph, it is too long and it has just one reference. The same happens for lines 58-70.

·      Please, remove from the end of the introduction the commercial names of the materials tested. They should appear in the M&Ms section.

·      In general, the introduction should be shrunk. It’s longer than the discussion section. Superfluous information should be avoided (e.g. type of grafts), focusing on the main topic of the article. It would be interesting, instead, adding few references on peri-implantitis and its management with adjuvant systems, as its one of the major concerns in this field: https://doi.org/10.3390/app12062800https://doi.org/10.1111/adj.12705).

Materials and Methods 

·      Approval from Ethics Committee or Institutional Review Board is missing. Please, provide the name of the IRB and the number of approval at the beginning of M&Ms section.

·      amoxicillin + clavulanic acid

·      0.20% chlorhexidine digluconate mouthwash

·      Which anesthetic was used? 

·      Materials should be presented in this way when they are cited for the first time: material (model/version, manufacturer, city, state). Then, if the manufacturer is the same also for other materials, you can just write material (manufacturer). 

·      Please, add the impression material (line 140).

Results

·      Please, clarify SLA acronym (line 172)

·      You can use only one table to present data. As they are presented now, they are confusing. Moreover, the headings should be more explanatory. 

·      Pay attention to the layout, figures are positioned in a messy way.

Discussion

The discussion section should be implemented. Several aspects were not taken into consideration. 

·      Home oral hygiene of patients were not considered, together or not with adjuvant systems. 

·      The use of drugs such as antibiotics could have influenced oral microbiome. 

·      The lack of a control group in which no porcine cortical bone lamina was not used is a great limitation of the study.

·      Other limitations are: the small sample size, the validity of the results only for the materials tested and the fact that a randomized clinical trial should have been performed to improve the reliability of the study. 

Conclusions

The limitations of the study here presented should be moved at the end of the discussion section and should be expanded as suggested. 

The conclusions should briefly present the results of the work with future perspectives. 

Authors disclosures

The article lacks: 

-       Author contributions

-       Funding

-       Institutional Review Board Statement

-       Informed consent statement

-       Data availability statement 

-       Conflicts of interest

Please, add the abovementioned sections, otherwise the article cannot be published. 

General issues

·      Informal writing is preferred, avoid using “our results”, “our case series”, and so on. Please, rephrase among the manuscript. 

·      The format of the citations in the text was not followed. The citation in the text should appear as “…” [1], and not “…” 1.

·      The format of the reference list was not observed. Please, refer to the Instruction for Authors and properly format all the references.

 .    English check from a native speaker is suitable as various grammatical and spelling issues were detected. 

Author Response

Response Reviewer 2: Dear Reviewer,

Thank you for your work in revising our manuscript. Your comments have allowed us to better clarify the suggested aspects, improving the quality of the manuscript. We corrected and reorganized the manuscript, figures and tables. As the corresponding author, I have provided a point-by-point response and highlighted in yellow the parts added in the text.

Title

I believe that the title should not contain “Case series”. This is a research article, but this should not be written in the title. 

Response: Thank you for your suggestions. We removed 'case series', but also reduced the title as suggested by another reviewer.

Keywords: Being Applied Sciences a multidisciplinary Journal, keywords like “dentistry” and “dental implants” should be added.

Response: Thank you. We added the suggested keywords.

Introduction

  • Lines 49-57: other references should be added in this paragraph, it is too long and it has just one reference. The same happens for lines 58-70.

Response: We have revised the introduction, corroborating the contents with appropriate references.

  • Please, remove from the end of the introduction the commercial names of the materials tested. They should appear in the M&Ms section.

Response: We removed commercial names from the introduction, specifying them in Mat&Met (lines 78-80).

  • In general, the introduction should be shrunk. It’s longer than the discussion section. Superfluous information should be avoided (e.g. type of grafts), focusing on the main topic of the article.

Response: We have modified the introduction, reducing, as recommended, the discussion on grafting materials, thus leaving a necessary part because the study concerns the use of particular grafts.

It would be interesting, instead, adding few references on peri-implantitis and its management with adjuvant systems, as its one of the major concerns in this field: https://doi.org/10.3390/app12062800https://doi.org/10.1111/adj.12705).

In this regard, we received specific instructions from the Academic Editor not to add these topics and suggested articles.

Materials and Methods 

  • Approval from Ethics Committee or Institutional Review Board is missing. Please, provide the name of the IRB and the number of approval at the beginning of M&Ms section .

Response: We have added the Ethics Committee and the approval number to the beginning paragraph of the M&Ms section (99-100).

  • amoxicillin + clavulanic acid

Response: We have modified as suggested (line 116).

  • 0.20% chlorhexidine digluconate mouthwash

Response: We have modified as suggested (line 117).

  • Which anesthetic was used? 

Response: We have added and specified (lines 119-120).

  • Materials should be presented in this way when they are cited for the first time: material (model/version, manufacturer, city, state). Then, if the manufacturer is the same also for other materials, you can just write material (manufacturer). 

Response: We have modified, specifying each material cited.

  • Please, add the impression material (line 140).

Response: We have specified the impression materials used (lines 155-156).

Results

  • Please, clarify SLA acronym (line 172)

Response: We have specified SLA acronym (line 215).

  • You can use only one table to present data. As they are presented now, they are confusing. Moreover, the headings should be more explanatory. 

Response: We have made the headings should be more explanatory and rearranged the tables more clearly. Having different values in comparison, we could not merge them.

  • Pay attention to the layout, figures are positioned in a messy way.

Response: We have reorganised the layout of the figures.

Discussion

The discussion section should be implemented. Several aspects were not taken into consideration. 

  • Home oral hygiene of patients were not considered, together or not with adjuvant systems. 

Response: This is certainly an important aspect why we also added the clinical practice of maintaining oral hygiene in Mat & Met, which also made the sample more homogeneous for observation.

  • The use of drugs such as antibiotics could have influenced oral microbiome. 

Response: Another very important aspect, which would require extensive discussion. We have listed the bibliographical references of the clinical procedures adopted.

  • The lack of a control group in which no porcine cortical bone lamina was not used is a great limitation of the study.

Response: We have specified this limit, as suggested.

  • Other limitations are: the small sample size, the validity of the results only for the materials tested and the fact that a randomized clinical trial should have been performed to improve the reliability of the study. 

Response: Thank you for your comments. We have specified these missing elements as limitations of the study.

Conclusions

The limitations of the study here presented should be moved at the end of the discussion section and should be expanded as suggested. 

The conclusions should briefly present the results of the work with future perspectives. 

Response Thank you! As suggested, we have moved the parts on limitations in the discussions, briefly added the results and future perspectives. 

Authors disclosures

The article lacks: 

-       Author contributions

-       Funding

-       Institutional Review Board Statement

-       Informed consent statement

-       Data availability statement 

-       Conflicts of interest

Please, add the abovementioned sections, otherwise the article cannot be published. 

Response: Thank you for your observation. We have added these disclaimers, not previously uploaded.

General issues

  • Informal writing is preferred, avoid using “our results”, “our case series”, and so on. Please, rephrase among the manuscript. 

Response: We have eliminated these locutions and modified.

  • The format of the citations in the text was not followed. The citation in the text should appear as “…” [1], and not “…” 1.

Response: As suggested, we have changed the format of the citations in then text, following the journal's guidelines and using Zotero software.

  • The format of the reference list was not observed. Please, refer to the Instruction for Authors and properly format all the references.

Response: As suggested, we have changed the format of the references list, following the journal guidelines and using Zotero software.

 .    English check from a native speaker is suitable as various grammatical and spelling issues were detected. 

Response: We corrected it, considering the journal services in the revision process.

Reviewer 3 Report

In this manuscript the authors have studied porcine cortical bone lamina for guided bone regeneration for dental implant treatment application. The authors have evaluated 30 dental implants in 20 patients for the radiographic bone loss in 24 months and have performed histomorphometric analysis on biopsy tissue after 8 months. The authors have concluded that the studied method represents a successful method to restore bone defects.

We thank the authors for this work that apparently required huge efforts to perform over a span of two years. However, there is an area to enhance and improve the current manuscript to proceed further for publication, as it would not be publishable suggesting the authors to re-submit after considering these points.

The main major points that were taken to reach this decision are:

Line 95: the inclusion criteria is missing a very important aspect, the missing units. When describing the patients with partially edentulous the authors have not mentioned how many missing teeth and how many implants per patient.

There are no exclusion criteria

Given that the authors have undertaken periodontal treatment prior to the surgical operation, what was the condition of the patients? i.e. have the authors considered periodontal issues in the selection criteria? The results may have a bias if healthy patients were to be compared with patient who had a history of periodontal disease, despite those patients have been treated prior to the surgery.

Line 142: How many assessors have assessed the measurements on the radiographs?  Were they blinded? if so, have the authors calibrated the assessors? Or was it one assessor ?

Line 178: The authors have stated “we observed a slight resorption” was this resorption significate? How the authors would interpretate slight ? It does not seem that the authors have done statistical analysis

Peri-implant bone levels assessment line 151, the authors have not explained how the mesial/distal bone margins were determined ? neither this is marked on the figure. I would suggest adding the point of measurement on the figures.

How the authors ensured the measure to the same point of the bone margin after two years ?

As per the histologic and histomorphometric results, how many slides were assessed ? what were the locations of the slides taken for quantification ?

It would be impossible to judge the total percentage of newly formed bone in the regenerated area in the socket, from merely quantification of some histologic slides obtained from the biopsy. This results in bias, hence the histomorphometric analysis results is not clear and needs further explanations.

Line 164: The authors have not given details about ImageJ analysis, as this software is a general use software and the authors have to follow special image segmentation protocols to obtain the area measurements in percentage ? Also, there is no mention about the total area of the slide that was included in the quantification measurements

The authors only present descriptive statistical analysis, no details about any comparative statistical analysis. This study lacks proper statistical analysis that would help in results interpretation and further backs authors’ conclusions. Proper statistical comparisons are needed to be included in the study

The authors are comparing single unit implant supported prosthesis with 3-unit bridge implant supported prosthesis this makes the results a bit biased. Furthermore, the authors have not given the percentages or numbers of each type of prosthesis in the included patients

Also, I include some other minor issues :

Line 85: “provide an histological evaluation”, typo “provide a histological evaluation”

Have the authors obtained an ethical approval to perform the study ? if so please indicate that in the text and add the approval reference number

Pre-surgical phase: I would assume that the authors have taken baseline pre-operative radiographs are well.

Line 138: Details about abutments used in this study is needed

Figures

Figure 5: add the mesial/distal bone margins measurement points

Figures 6-7: add scalebar, highlight features. Add segmentation image

Lines 289-292: The last paragraph in the conclusion does not seem to belong to the manuscript!

Author Response

Dear Reviewer,

Thank you for your work in revising our manuscript. Your comments have allowed us to better clarify the suggested aspects, improving the quality of the manuscript. We corrected and reorganized the manuscript, figures and tables. As the corresponding author, I have provided a point-by-point response and highlighted in light blue the parts added in the text.

The main major points that were taken to reach this decision are:

Line 95: the inclusion criteria is missing a very important aspect, the missing units. When describing the patients with partially edentulous the authors have not mentioned how many missing teeth and how many implants per patient.

Response: As suggested, we have reported the range of missing elements for each patient we have included, rehabilitated with 1 or 2 implants, as later reported. As this is a retrospectively analyzed case series, we included patients who were treated consecutively in the clinic with posterior mandibular partial edentulism (lines 87-89).

There are no exclusion criteria

Response: We have added the exclusion criteria used in our clinic for these interventions (lines 90-98).

Given that the authors have undertaken periodontal treatment prior to the surgical operation, what was the condition of the patients? i.e. have the authors considered periodontal issues in the selection criteria? The results may have a bias if healthy patients were to be compared with patient who had a history of periodontal disease, despite those patients have been treated prior to the surgery.

Response: Thank you for your request for clarification. The periodontal treatment was performed in the 12 months prior to GBR surgery and only periodontally stabilized patients were included. We have reported it in the text (lines 103-106).

Line 142: How many assessors have assessed the measurements on the radiographs?  Were they blinded? if so, have the authors calibrated the assessors? Or was it one assessor ?

            Response: Thank you. We have specified what is required (lines 166-169)

Line 178: The authors have stated “we observed a slight resorption” was this resorption significate? How the authors would interpretate slight ? It does not seem that the authors have done statistical analysis

Response:  we have eliminated “slight” (line 224); Since this is a case series, a descriptive statistic, reporting a pre-post difference for MBL, provides sufficient information, whereas a comparative statistic in the absence of a control group not indicated, as previously published guidelines have indicated (Grimes DA, Schulz KF. Descriptive studies: what they can and cannot do. Lancet. 2002 Jan 12;359(9301):145-9. doi: 10.1016/S0140-6736(02)07373-7. PMID: 11809274). The data are shown in the table, and we have reported average values in the results.

Peri-implant bone levels assessment line 151, the authors have not explained how the mesial/distal bone margins were determined ? neither this is marked on the figure. I would suggest adding the point of measurement on the figures.

Response: We have better specified the measurement procedure (lines 174-178) and added elements to the image.

How the authors ensured the measure to the same point of the bone margin after two years ?

Response: To ensure measurement of the same bone margin point after two years, silicone controls were used on the biteblocks, then stored properly and used for the second measurement (lines 170-172).

As per the histologic and histomorphometric results, how many slides were assessed ? what were the locations of the slides taken for quantification ?

Response: We have added this information (lines 188-190).

It would be impossible to judge the total percentage of newly formed bone in the regenerated area in the socket, from merely quantification of some histologic slides obtained from the biopsy. This results in bias, hence the histomorphometric analysis results is not clear and needs further explanations.

Response: Thank you for your note. The procedure certainly cannot represent the total percentage of newly formed bone, but in line with previous studies in the literature (e.g. those mentioned 24 and 25), it is a way to get information on the graft integration processes.

Line 164: The authors have not given details about ImageJ analysis, as this software is a general use software and the authors have to follow special image segmentation protocols to obtain the area measurements in percentage? Also, there is no mention about the total area of the slide that was included in the quantification measurements

Response: We have added image analysis. The different areas were measured in pixels, and the total area is a necessary measure to calculate the percentage of the different components analyzed. So, there are 20 areas in pixels for each analyzed slice. This procedural data, in our opinion, could be misleading and not very useful to include. We have reported the procedure performed for image segmentation (lines 200-210).

The authors only present descriptive statistical analysis, no details about any comparative statistical analysis. This study lacks proper statistical analysis that would help in results interpretation and further backs authors’ conclusions. Proper statistical comparisons are needed to be included in the study

Response: Thank you for your comment. Aware of the limitations of this study design, as this is a case series, a descriptive statistic provides sufficient information, whereas a comparative statistic in the absence of a control group would be excessive, as previously published guidelines have already indicated (Grimes DA, Schulz KF. Descriptive studies: what they can and cannot do. Lancet. 2002 Jan 12;359(9301):145-9. doi: 10.1016/S0140-6736(02)07373-7. PMID: 11809274) and would not add statistical power. Also in the case of histomorphometry there is only one detection in the study, with no value to compare

The authors are comparing single unit implant supported prosthesis with 3-unit bridge implant supported prosthesis this makes the results a bit biased. Furthermore, the authors have not given the percentages or numbers of each type of prosthesis in the included patients

Response: Thank you for the clarification request; we have reported the percentages of the different prosthetic structures (lines 216-218). We agree that this can make the results a bit biased, but this is a retrospective evaluation, and we have reported the different types of prosthesis found. In a prospective RCT it would be desirable to have a homogeneous sample for the prosthetic structures as well, to reduce procedural bias.

Also, I include some other minor issues :

Line 85: “provide an histological evaluation”, typo “provide a histological evaluation”

            Response: We have corrected it (line 76)

Have the authors obtained an ethical approval to perform the study ? if so please indicate that in the text and add the approval reference number

            Response: We have reported it (lines 358-361)

Pre-surgical phase: I would assume that the authors have taken baseline pre-operative radiographs are well.

Response: Thank you for your observation. Yes, and we have reported it in the text. (lines 110-111)

Line 138: Details about abutments used in this study is needed.

Response: We have reported it (lines 157-159)

Figures

Figure 5: add the mesial/distal bone margins measurement points

            Response: The points are indicated by arrows in red. We have specified this in the caption

and added explanations.

Figures 6-7: add scalebar, highlight features. Add segmentation image

            Response: We have added this information We added the image as an appendix and did not insert it before the histological images, maintaining the consequentiality of the writing. (lines 371-374).

Lines 289-292: The last paragraph in the conclusion does not seem to belong to the manuscript!

            Response: We have eliminated it.

Reviewer 4 Report

This article is a case series presenting a technique for guided bone regeneration. The topic of the article is very interesting, albeit the paper has many flaws that need to be addressed before publication.

Regarding the Title

Please consider a slight simplification of the title to make it more readable for readers otherwise I consider it suitable

Also please remove the sentence underlines 18 and 19.

Regarding the Abstract

Please remove the headlines from the abstract (background methods results conclusion). also please improve the English syntax of the abstract.

Regarding the Keywords

Cancel the bold style on the keywords

Regarding the Introduction

References to registered trademarks on the lines 88 - 91 should be properly referenced, including the headquarters /city and the country. This chapter contains many grammatical errors, please correct.

In this chapter Authors reference in the line 77 two publications including one auto citation that shall be referencing reverse colorization. Not only these references are not in proper format.

Regarding the Materials and methods

This chapter describes a sample of 20 patients admitted to oral surgery unit in the year 2016. These patients were treated with guided bone regeneration with the application of porcine heterologous cortical bone laminae. This case series is very similar to our published case series or the same author from the year 2016. The author has previously published in June 2016 a publication “The use of resorbable cortical lamina and micronized collagenated bone in the regeneration of atrophic crestal ridges: a surgical technique. Case series Article in Journal of Biological Regulators and Homeostatic Agents · June 2016”,. The author has performed this published research also on the sample of 20 patients with also and 30 implants were inserted. Authors shall explain how this paper is different from the paper published in 2016 and this shall be disclosed prior submission.

The article has a very important part removed from the end including author contributions funding and most importantly Institutional Review Board statement and informed consent statement.

The study is performed on the patients and includes clinical intervention and evaluation. These must be approved by patient written informed consent and also approved by institutional review board.

Regarding the list of literature

This list is incorrect, there are duplicate numberings of references which many are in the wrong format.

Summary

I started reading this article with high expectations, as the topic is very interesting. However, the similarities to a previously published paper with possible republication, the auto-plagiarism of a previously published patient sample, and the omission of important legal aspects such as informed patient consent and institutional advisory board opinion, as well as many errors in the text and improper formatting and referencing, make this paper unsuitable for publication in my view.

Author Response

Dear Reviewer,

Thank you for your work in revising our manuscript. Your comments have allowed us to better clarify the suggested aspects, improving the quality of the manuscript. We corrected and reorganized the manuscript, figures and tables. As the corresponding author, I have provided a point-by-point response and highlighted in green the parts added in the text.

Regarding the Title

Please consider a slight simplification of the title to make it more readable for readers otherwise I consider it suitable

Response: Thank you for your advice. As suggested, we have simplified the title to make it more readable.

Also please remove the sentence underlines 18 and 19.

            Response: Thank you! As suggested, we have removed the sentence.

Regarding the Abstract

Please remove the headlines from the abstract (background methods results conclusion). also please improve the English syntax of the abstract.

Response: We removed the headlines from the abstract. We are reviewing English syntax possibly considering the journal services.

Regarding the Keywords 

Cancel the bold style on the keywords

Response: We have corrected as suggested.

Regarding the Introduction

References to registered trademarks on the lines 88 - 91 should be properly referenced, including the headquarters /city and the country. This chapter contains many grammatical errors, please correct.

Response: We have removed the raw material specifications from the introduction, as suggested by another reviewer, and reported them correctly in Mat&Met. In addition, we checked the English correction, considering the journal services.

In this chapter Authors reference in the line 77 two publications including one auto citation that shall be referencing reverse colorization. Not only these references are not in proper format.

            Response: We have modified the introduction in some parts.

Regarding the Materials and methods

This chapter describes a sample of 20 patients admitted to oral surgery unit in the year 2016. These patients were treated with guided bone regeneration with the application of porcine heterologous cortical bone laminae. This case series is very similar to our published case series or the same author from the year 2016. The author has previously published in June 2016 a publication “The use of resorbable cortical lamina and micronized collagenated bone in the regeneration of atrophic crestal ridges: a surgical technique. Case series Article in Journal of Biological Regulators and Homeostatic Agents · June 2016”, The author has performed this published research also on the sample of 20 patients with also and 30 implants were inserted. Authors shall explain how this paper is different from the paper published in 2016 and this shall be disclosed prior submission.

Response: Dear Reviewer,

The study cited is a specification of the surgical technique and relates to a broader research programme, in which several materials were tested, the pilot evaluations of which could have directed an RCT. Because of this, the periods and case numbers oare similar so that they can be later compared, but the materials are different. In the aforementioned study, a micronised particulate (particle size up to 300 µm) was used, as also specified in the title, which revealed faster resorption, whereas in this study, a filler with a particle size of 600-1000 µm was used.

Having certainly clarified your misunderstanding, we will also elucidate your remark to the Editors.

Best Regards

The article has a very important part removed from the end including author contributions funding and most importantly Institutional Review Board statement and informed consent statement.

Response:  We Inserted these parts, which were not uploaded in the initial manuscript.

The study is performed on the patients and includes clinical intervention and evaluation. These must be approved by patient written informed consent and also approved by institutional review board.

Response:  We specified these important parts, which were not uploaded in the initial manuscript, in the respective entries

Regarding the list of literature

This list is incorrect, there are duplicate numberings of references which many are in the wrong format.

Response: We have corrected and edited the references.

Summary

I started reading this article with high expectations, as the topic is very interesting. However, the similarities to a previously published paper with possible republication, the auto-plagiarism of a previously published patient sample, and the omission of important legal aspects such as informed patient consent and institutional advisory board opinion, as well as many errors in the text and improper formatting and referencing, make this paper unsuitable for publication in my view.

Response: Dear Reviewer,

We specified the similarities you found with another study in the previous note on Materials & Methods, we have reported disclaimers not previously uploaded regarding informed patient consent and the Ethics Committee statement, corrected errors in the text and reformatted the references.

Best Regards

Round 2

Reviewer 2 Report

Dear Authors, 

Thank you for providing the revised version of your manuscript. 

Two points were not addressed: 

- the addition of references on peri-implantitis and its management with adjuvant systems (https://doi.org/10.3390/app12062800; https://doi.org/10.1111/adj.12705)

- the rephrasing of personal writing ("we", "our").

Once performed the modifications, the manuscript will be suitable for publication.

Author Response

Dear author,

thank you for your further requests for clarification, which will improve the study. I have provided a point-by-point response and keeping track of it in the text.

Two points were not addressed: 

- the rephrasing of personal writing ("we", "our").

Response: We modified sentences with this construct (line 212, 224, 281, 294).

- the addition of references on peri-implantitis and its management with adjuvant systems (https://doi.org/10.3390/app12062800; https://doi.org/10.1111/adj.12705)

Response:  Dear author,

regarding this suggestion, as reported in the previous revision responses, I received a message from the Academic Editor with specific instructions not to follow this suggestion, and therefore not to broach the topic and not to cite the two studies you suggested.

In any case, the Academic Editor can give you further explanation.

Certain that we have clarified the point, we thank you for your work and time spent on the revision and we hope that the manuscript is now suitable for publication.

Best regards

Reviewer 4 Report

Dear authors,

Thank you for responses to my previous remarks.

Also, for adding the Ethics Committee statement - Institutional Review Board Statement was issued this year? Not prior to the described research? ("Fondazione Policlinico Universitario A. Gemelli - #0009738/22”.)

Thank you for explaining the difference between your former 
very similar research from June 2016. “The use of resorbable cortical lamina and micronized collagenated bone in the regeneration of atrophic crestal ridges: a surgical technique. Case series Article in Journal of Biological Regulators and Homeostatic Agents ”. It shall also be explained well in the text differences between these two, I might have overlooked it in the new revision, as there are many substantial changes. It may be confusing for many as the sample of 20 patients with also and 30 implants match this paper and as you explain only the material is different. This research must be referenced as well and all similar papers published by it. A significant portion of this research ich shared with this paper.

I leave this up to the editors to evaluate if this is the case of auto-plagiarism/self-plagiarism. Now I believe it is not upon your explanation. As this is not the case of authors reusing significant portions of their previously published work without attribution. 
I believe this paper is clinically useful and can be published.

Author Response

Dear author,

thank you for your further requests for clarification, which will improve the study. I have provided a point-by-point response and keeping track of it in the text.

1) Also, for adding the Ethics Committee statement - Institutional Review Board Statement was issued this year? Not prior to the described research? ("Fondazione Policlinico Universitario A. Gemelli - #0009738/22”.)

Response: Dear Reviewer,

The date of approval by the Ethics Committee concerns our request to retrospectively reanalyze existing data from interventions performed in 2016, as specified at the beginning of Mat&Met, lines 84-85.

2) Thank you for explaining the difference between your former very similar research from June 2016. “The use of resorbable cortical lamina and micronized collagenated bone in the regeneration of atrophic crestal ridges: a surgical technique. Case series Article in Journal of Biological Regulators and Homeostatic Agents ”. It shall also be explained well in the text differences between these two, I might have overlooked it in the new revision, as there are many substantial changes. It may be confusing for many as the sample of 20 patients with also and 30 implants match this paper and as you explain only the material is different. This research must be referenced as well and all similar papers published by it. A significant portion of this research ich shared with this paper.

Response: Dear author,

as recommended, we have added in the discussion an additional specification related to the other cited study (284-285). In the text: “Some authors of this study used the same protocol in a parallel case study, except for micronized filling (200-300 µm), but revealed faster resorption and reduced graft stability”.

3) I leave this up to the editors to evaluate if this is the case of auto-plagiarism/self-plagiarism. Now I believe it is not upon your explanation. As this is not the case of authors reusing significant portions of their previously published work without attribution. 
I believe this paper is clinically useful and can be published.

Response: Dear author,

Thank you for your request to clarify this point. I have already informed and resolved with the editors on this question.

Dear author,

We thank you for your work and time spent on the revision. We hope that the manuscript is now suitable for publication.

Best Regards

Round 3

Reviewer 2 Report

Dear Authors, 

Thank you for your response. The manuscript is now suitable for publication. 

Author Response

Dear reviewer,

we are extremely grateful for your review work.

Best regards